# A Perspective on the Development of c-Jun N-terminal Kinase Inhibitors as Therapeutics for Alzheimer’s Disease: Investigating Structure through Docking Studies

**DOI:** 10.3390/biomedicines9101431

**Published:** 2021-10-09

**Authors:** Hyunwook Cho, Jung-Mi Hah

**Affiliations:** 1Department of Pharmacy, College of Pharmacy, Hanyang University, Ansan 15588, Korea; lod0201@hanyang.ac.kr; 2Center for Proteinopathy, Institute of Pharmaceutical Science and Technology, Hanyang University, Ansan 15588, Korea

**Keywords:** Alzheimer’s disease, c-Jun N-terminal kinase, small-molecule protein kinase inhibitor, JNK3, selectivity

## Abstract

c-Jun N-terminal kinase (JNK) plays an important role in cell death caused by various stimuli. Because the isoform JNK3 is mainly expressed in the brain, it is believed to play a pivotal role in various neurodegenerative diseases, including Alzheimer’s disease (AD) and Parkinson’s disease (PD), which still lack plausible therapeutics. To develop a novel and selective JNK3 inhibitor, we conducted a decadal review (2011 to 2021) of published articles on JNK inhibitors, particularly those focusing on a structural perspective and docking insights. We observed the structures of three isoforms of JNK, namely holo-proteins and co-crystal structures, with JNK3 inhibitors and summarized the significant structural aspects of selective JNK3 inhibitors as AD therapeutics.

## 1. Introduction

Protein kinases were discovered more than 65 years ago, but they have been therapeutic targets for less than 30 years [1,2]. The involvement of protein kinases in many diseases has been established based on a large number of studies, and such kinases have long been a promising molecular target group. There are 518 known protein kinases encoded in the human genome, and protein kinases phosphorylate approximately one-third of all proteins [3,4]. Many researchers in pharmaceutical companies and academia have worked hard in this area, and as a result, 67 small-molecule kinase inhibitor drugs have been approved by the US FDA as of 2021 [5].

Gleevec, a well-established treatment for chronic myeloid leukemia, was launched in 2001 after being developed by Novartis, initiating the protein kinase inhibitor era. This treatment is still extending its range of applications. Following this, nine small-molecule protein kinase inhibitors were approved by the US FDA as of 2011, and they exclusively targeted proliferative diseases such as cancer. Similarly, most protein kinase inhibitors approved thus far are used in oncology. However, protein kinases are also involved in many other diseases, including inflammatory, immunological, and cardiovascular diseases, as well as central nervous system (CNS) disorders such as Alzheimer’s disease (AD) and Parkinson’s disease (PD) [6]. In 2011, ruxolitinib, which is a JAK1 and JAK2 inhibitor, was approved as a therapeutic agent for myelofibrosis. Along with this approval, research in the field of non-oncology kinase inhibitors became increasingly active [7]. There are as many as 10 drugs targeting diseases outside of oncology among the 67 FDA-approved protein kinase inhibitors, but there is no protein kinase inhibitor drug for CNS disorders yet.

AD was discovered in 1906, and a post-mortem examination observed two clear biomarkers, namely tau tangle and amyloid beta (Aβ) plaque, but as of 2021, a clear therapeutic target has not yet been identified. With the rapid aging of the global population, neurodegenerative diseases such as AD have become a major social issue and have a significant impact not only on patients, but also on other individuals around the patients [8,9]. Because small molecule protein kinase inhibitors are being used for a broad range of diseases, molecular targets have attracted significant attention in the area of CNS disorders. Out of many potential CNS protein kinases, we selected c-Jun N-terminal kinase (JNK), a member of the mitogen-activated protein kinase family. The JNKs phosphorylate multiple apoptosis-related transcriptional factors, including c-Jun, ATF, APP, and tau, and induce cellular apoptosis [10].

JNKs respond to various stimuli such as cytokines, neurotoxins, oxidative stress, and fatty acids. When stimuli first reach the cell membrane, signals are transferred to mitogen-activated protein kinase (MAPKKK), MAPKK4, and MAPKK7 through phosphorylation. Next, the activated MAPKK4 and MAPKK7 phosphorylate JNK at two different sites, and the phosphorylated JNK then phosphorylates the N-terminal of c-Jun to induce an apoptotic signal [11]. Additionally, JNK directly phosphorylates apoptotic proteins such as BIM and BMF and activates them to activate caspases [12]. Overall, JNKs are heavily involved in the physiological processes of apoptosis. There are three different human JNK genes, namely *jnk1*, *jnk2*, and *jnk3* that encode 10 different splice JNK variants (4 JNK1/2 homozygous and 2 JNK3 homologous), of which JNK1 and JNK2 are the most widely expressed (Figure 1).

Unlike JNK1 and JNK2, JNK3 is mostly expressed in the brain, and only a small portion is expressed in the heart and testis. JNK3 has been considered a potential therapeutic target for neurodegenerative diseases associated with neuronal cell death. When c-Jun is phosphorylated by JNK3, it also facilitates tau tangle maturation. Additionally, JNK3 is known to phosphorylate tau proteins directly, leading to the formation of tau tangles.

Another biomarker of AD pathology, namely amyloid beta, is also related to JNK3. JNK3 phosphorylates amyloid precursor protein (APP) at T688, causing APP endocytosis, which is known to be the most important step in the entire amyloid-beta formation process [13].

The mechanism underlying Aβ plaque-induced apoptosis remains unclear. Aβ plaques induce the phosphorylation of AMP-activated protein kinase (AMPK), and activated AMPK phosphorylates TSC2 and Raptor at S1387 and S792. This process inhibits the mTOR pathway and the representative cell rescue system and induces a translational block that prevents protein expression [14]. This phenomenon induces widespread oxidative endoplasmic reticulum stress, which is accompanied by an unfolded protein response that induces secondary reactions such as the inflammatory response, which reactivates JNK3. Another rationale for predicting that neuronal cell death will decrease in the brains of AD patients when JNK3 is inhibited is that neuronal cell death and cognitive decline were also reduced in a JNK3 knock-out FAD mouse model [15] (see Figure 2).

Over the past decade, JNK inhibitors have attracted attention as therapeutic targets for AD. However, only a few JNK inhibitors have been developed through clinical trials, and one of the main reasons is isoform selectivity. In this review, we will examine some pan-JNK inhibitors and selective JNK3 inhibitors that have been reported thus far and propose directions for future AD therapeutics. Additionally, we propose a novel design for selective JNK3 inhibitors by analyzing differences in crystal structures between JNK1, JNK2, and JNK3.

## 2. Current Development Status of JNK Inhibitors

### 2.1. Pan-JNK Inhibitors

SP600125 (Figure 3) is the first known pan-JNK inhibitor with IC_50_ values of 90, 40, and 40 nM for JNK3, JNK2, and JNK1, respectively. This compound was the first JNK inhibitor studied, and the understanding of the intracellular signaling pathway of JNK has been expanded. Initially, it was developed using cancer cells as indicators of apoptotic cell death, but later, this compound exhibited a neuroprotective effect in MPTP-induced PD animal models. It also reduced neurofibrillary entanglement and Aβ plaques in AD animal models [16,17,18]. The potency and selectivity of SP600125 did not lead to drug development, but it is a compound that has made many scientists consider JNK an attractive drug target.

AS602801 (Figure 3) is an orally active pan-JNK inhibitor. IC_50_ values of 230, 90, and 80 nM are exhibited for JNK3, JNK2, and JNK1, respectively. It demonstrated a significant effect on endometriosis when it was used in combination therapy with hormones or as a monotherapy in baboon and rodent models. While AS602801 is not a selective inhibitor, there was an attempt to target immune-related diseases with AS602801 treatment, unlike SP600125. This compound entered phase II clinical trials in 2012 with endometriosis as an indication [19,20].

Tanzisertib (Figure 3) is the third orally active pan-JNK inhibitor, with IC_50_ values of 6, 7, and 61 nM for JNK3, JNK2, and JNK1, respectively. Tanzisertib was proven to be effective in animal experiments using a bleomycin-induced pulmonary fibrosis model as a target disease for idiopathic pulmonary fibrosis. Experiments have also demonstrated that it reduces the production of TNF-α in an acute rat LPS-induced TNF-α production PK-PD model. It entered phase II clinical trials with idiopathic pulmonary fibrosis as an indication, but the trials have been halted for unknown reasons [21].

### 2.2. Selective JNK Inhibitors

Compound **1** (Figure 4) is a selective JNK3 inhibitor with activity levels of <1 nM for JNK3, 210 nM for JNK2, and 518 nM for JNK1. Unlike the pan-JNK inhibitors, it is characterized by isoform selectivity toward JNK3, which is mainly expressed in the brain. Therefore, it was developed to target Parkinson’s disease [22]. Compound **2** (Figure 4) was synthesized from compound **1**. It has a thiophenyl pyrazolourea scaffold and selectivity for other kinases, as well as JNK1 and JNK2 isoforms. Its activity on JNK3 has an IC_50_ of 35 nM, and it exhibits a much better DMPK profile in vivo compared to previously reported JNK3 inhibitors (**1** in Figure 4). It has also been demonstrated that this compound could be orally available and blood brain barrier (BBB)-penetrable [23].

Compound **3** (Figure 4) was discovered through multistage structure-based virtual screening. It exhibits a 40 nM IC_50_ for JNK3 and 2500-fold isoform selectivity over JNK1 and JNK2, as well as a good selectivity profile for 398 kinases. It also exhibits greater neuroprotective activity against Aβ-induced cellular toxicity in SH-SY5Y cells compared to LiCl, which is a potential therapeutic agent for AD in vitro. Animal studies using mice revealed increased learning and memory abilities, and studies of the underlying mechanisms revealed that this compound can significantly reduce the diffusion of fibrillar Aβ plaques through the inhibition of JNK following the reduced phosphorylation of APP and tau proteins in the cortex and hippocampus. [24].

Compound **4** (Figure 4) is a JNK2- and JNK3-selective inhibitor with IC_50_ values of 16 nM for JNK3, 97 nM for JNK2, and 420 nM for JNK1. JNK2 isoform selectivity was not achieved, but the compound did exhibit high selectivity over other MAPK families and JNK1 isoforms. As a JNK3 inhibitor for neurodegenerative disease therapeutics, this compound exhibited favorable BBB permeability [25].

## 3. Structural Perspective Analysis of JNK’s Active Site

### 3.1. Superimposition of JNK1, JNK2, and JNK3 Crystal Structures

To compare the structures of JNK1 and JNK3, compound **3** was docked into JNK3 (Protein Data Bank (PDB) ID: 4WHZ). The indolin-2-one component is located in the hinge region and interacts with the hydrogen of NH and carbonyl oxygen in the backbone of Met149. The nitrogen of thiazol-4-one forms a hydrogen bond with Lys93 through the water bridge, and the 2-chloro-phenyl group is buried in the hydrophobic pocket consisting of Ile124, Leu126, Leu144, Val145, Met146, and Leu206, which is located on the backside of the active site beyond the gatekeeper residue Met146. Met146 interacts with an aromatic ring (in compound **3**, the 2-chloro-phenyl group) in JNK3.

When the docked pose of compound **3** with JNK3 was superimposed onto the crystal structures of JNK1 (PDB ID: 3PZE, 2XRW, 4QTD, 3ELJ, 4AWI, and 4L7F) (Figure 5), it seemed that every Met108 residue in the observed JNK1 structures could conflict with the 2-chloro-phenyl group occupying the selectivity pocket, in contrast to the methionine146 of JNK3 [26]. This methionine conflict could explain the selectivity of compound **3** for JNK3 over JNK1.

To determine how the gatekeeper residue methionine108 acts in the active site of JNK2, the docked pose of compound **3** was superimposed onto three published crystal structures of JNK2 (PDB ID: 3E7O, 7CML, and 3NPC) (Figure 6). Although the Met108 in 3E7O conflicts with the 2-chloro-phenyl group occupying the selectivity pocket, the Met108 from 7CML and 3NPC interacts with the 2-chloro-phenyl group through sulfur–pi interactions, similar to Met146 in the active site of JNK3 [32]. Either the two structures forming sulfur–pi interactions are holoenzymes without co-crystallized ligands, or the co-crystallized ligands already have aromatic rings occupying the selectivity pocket. In the crystal structure of compound **4**, which has isoform selectivity only for JNK1, the naphthalene moiety has a sulphur-pi interaction with the Met146 residue of JNK3 and the IC_50_ of JNK2 is 97 nM, which is a reasonable activity level. In the case of tanzisertib, considering the crystal structure of tanzisertib on JNK3 (not shown), the trifluorophenyl group occupies the selectivity pocket and interacts with Met146, but the activity of JNK2 is almost the same as that of JNK3. By combining these observations, we concluded that compound **4** induces the gatekeeper residue Met108 of JNK2 to move sufficiently to form a hydrophobic pocket similar to that of JNK3.

The behavior of the gatekeeper residue Methionine146 in the active site of JNK3 was observed using the same method described above (Figure 7). The Met146 groups in the four crystal structures of JNK3 (PDB ID: 4WHZ, 4W4W, 3OY1, 7KSI) exhibit sulfur–pi interactions with the 2-chloro-phenyl groups, excluding two crystal structures (PDB ID: 6EMH, 6EQ9) in which co-crystallized ligands do not occupy the hydrophobic pocket [26,27,28,29,30,31,32]. The compound-induced movement of methionine residues in the JNK2 structure and the formation of a hydrophobic pocket are more common in JNK3 structures. This tendency could explain the isoform selectivity of JNKs. None of the Met108 residues in the observed crystal structures of JNK1 move sufficiently to form a hydrophobic pocket, but in the crystal structures of JNK2 and JNK3, the methionine residues move sufficiently to create hydrophobic pockets with the inhibitors. Therefore, selectivity for JNK1 can be achieved through hydrophobic pocket occupancy, but the selectivity of JNK2 requires additional explanation.

### 3.2. Different Residues between JNK1 and JNK3

To investigate the residues comprising the active site of JNK3, we observed residue sequences 45 to 400 of JNK3 and found that the Leu144 residue exists in the active site as an element in the selectivity pocket. The Leu144 forms hydrophobic interactions with the naphthalene ring of compound **4** in the crystal structure of JNK3 (PDB ID: 3OY1) (Figure 8a). However, when the crystal structure of JNK1 (PDB ID: 3PZE) is superimposed onto the co-crystallized ligand of 3OY1 (Figure 8b), Ile106 exists in place of the Leu144 residue in the same location and conflicts with the naphthalene ring of compound **4** [37]. The distance between Leu144 in the crystal structure of JNK3 and the naphthalene ring is maintained at 3.53 Å, but Ile106 from the crystal structure of JNK1 causes bad contact with the naphthalene ring because it is too close at a distance of 2.26 Å. Because JNK2 also has Leu144 in the same location as JNK3, we can conclude that occupying the selectivity pocket is not an absolute condition to achieve selectivity for JNK1 and JNK2.

## 4. Functional Groups in Each Compound Contributing to Isoform Selectivity

To determine the structural clues for the selective inhibitor compound **1** in the binding mode, we analyzed the compound 1-JNK3 co-crystal complex in detail (Figure 9). A pair of hydrogen bonds is formed in the Met149 hinge region, the NH group of benzamide acts as a hydrogen bond donor, and the 2-nitrogen in the pyrazole ring acts as a hydrogen bond acceptor. Another hydrogen bond is formed between the urea group and Lys93 through a water bridge, and the 2-chloro-phenyl group is located deep in the hydrophobic pocket, forming a sulphur-pi interaction with the Met146 gatekeeper residue. The introduction of ortho-substituted chloride seems to be important in terms of selectivity, because it changes the angle of the two rings in terms of conformation and causes the phenyl group to fit into the selectivity pocket [22]. The protonated pyrrolidine in the solvent exposure area forms a hydrogen bond with Asn89 in the upper part of the active site. Compound **1** exhibits 210-fold selectivity for JNK3 over JNK2, which is much higher than that of the other selective JNK3 inhibitors examined. The hydrogen bond between pyrrolidine and Asn89 is the most reasonable explanation for this phenomenon.

A similar observation was made for the compound **2**-JNK3 co-crystal structure (Figure 10). Compound **2** also has two hydrogen bonds with Met149 in the hinge region, formed by the NH group from the amide and the 2-nitrogen of the pyrazole ring. An additional hydrogen bond was found in the urea moiety with Lys93 through the water bridge. The urea-connected 2-chloro phenyl group exhibited a sulfur–pi interaction with the Met146 gatekeeper residue within the selectivity pocket, and the *ortho* chloride substitution made this interaction more favorable. All of these interactions were very similar to the relationship of compound **1** with JNK3. The only difference compared to the compound **1** binding mode was the loss of hydrogen bonds in the solvent exposure component, which appears to be the reason for the change in JNK3 activity (from <1 nM to 35 nM) and the decrease in selectivity compared to JNK2 (from 210- to 39-fold) [23]. However, compound **2** showed a superior pharmacokinetic profile in vivo compared to compound **1**. Regardless, we believe that it is advantageous to maintain the hydrogen bond donor property in the solvent exposure component for the future design of selective JNK3 inhibitors.

The binding mode of compound **3** was examined to explore the rationale for compound **3** exhibiting 2500-fold selectivity over JNK2 (Figure 11). A co-crystal structure was not available for compound **3,** and we monitored the binding mode through docking simulations. As shown in Figure 5, the indolin-2-one component is located in the hinge region, where there are two hydrogen bonds with Met149, one hydrogen bond with Lys93 through the water bridge, and the hydrophobic interaction of the 2-chloro-phenyl group [24]. In addition to these interactions, we found that the 5-fluorine of indolin-2-one interacts closely with the Asn152 and Gln155 residues (distances of 4.69 Å and 3.41 Å, respectively) through F-bonding. These interactions are clear based on the differences in activity with and without fluorine substitution. Additionally, the side chain of Asn152 exhibits a dipole interaction with the carbonyl group of thiazol-4-one. Both the F-bond and the dipole interactions could explain the high selectivity of compound **3** for JNK2.

In contrast to the selectivity of compound **3**, we examined the low selectivity of compound **4** to identify its binding mode (Figure 12). It has two hydrogen bonds in the hinge region, and the carbonyl group of triazol-3-one directly forms a hydrogen bond with Lys93, unlike in the previous compounds. The naphthalene ring enters the selectivity pocket to form hydrophobic and sulfur–pi interactions with the residues described above [25]. Although the cyclohexane group was substituted in the solvent exposure component, no additional interactions were observed. In conclusion, compounds with selectivity for JNK2 commonly exhibit additional interactions in the form of either hydrogen bonds, F-bonds, or dipole interactions in the solvent exposure area.

## 5. Conclusions

Neurodegenerative diseases such as AD are becoming increasingly critical social problems based on the rapidly aging global population. AD is a disease with huge unmet therapeutic needs that requires new therapeutic molecular targets. Regarding new therapeutic targets, we investigated c-Jun N-terminal-kinase 3, which is a pivotal protein kinase in the neuronal apoptotic process. JNK plays an important role in the apoptotic process at the terminal of the MAPK pathway, and the isoform JNK3 is mainly expressed in the brain. Therefore, it is considered to be a target for neurodegenerative diseases, in which neuronal apoptosis is a key event. Additionally, because JNK3 appears repeatedly in several pathological pathways of AD, we considered it a promising target for overcoming this disease. 

Therefore, a comprehensive list of current pan-JNK inhibitors and selective JNK3 inhibitors was assembled, and their binding modes were examined. Selective JNK3 inhibitors commonly occupy a hydrophobic region (Ile124, Leu126, Leu144, Val145, Met146, and Leu206) called the selectivity pocket by introducing an aromatic ring into the JNK3 inhibitor so that it can enter the hydrophobic pocket deep in the active site of JNK3. Selectivity for the JNK1 isoform was achieved by allowing hydrophobic interactions and sulfur–pi interactions. Six JNK1, three JNK2, and six JNK3 crystal structures were selected according to their resolutions to compare the ligand-induced movements of gatekeeper Met residues. In each of the six JNK1 co-crystals, the methionine collided with the aromatic ring in the selective JNK3 inhibitor, and none of the co-crystallized ligands could fit into the hydrophobic pocket. The collision of the gatekeeper methionine with the aromatic ring was also observed in the co-crystal of JNK2, but if the co-crystallized ligand occupies the hydrophobic region, then the methionine of JNK2 is induced by the ligand and a pocket can be formed. 

Among the six observed inhibitor-JNK3 co-crystal structures, there were more cases with ligands occupying the hydrophobic pocket governing this selectivity. Four co-crystallized ligands occupied the hydrophobic pocket and two did not occupy the hydrophobic pocket. Although different occupancy levels were observed depending on the shapes of the four ligands, all methionines formed a hydrophobic pocket. 

As a supplementary explanation for the residues comprising the hydrophobic pocket deep in the active site, there is a difference between the Leu of JNK3 and the Ile of JNK1. Unlike the relatively short Leu, the Ile residue of JNK1 prevents the medium to large aromatic rings from entering the hydrophobic pocket. 

Because the leucine residue is located at the hydrophobic pocket of JNK2 as well as JNK3, the aromatic ring in the hydrophobic region has no significant impact on selectivity over JNK2. According to these results, one can see that additional elements are required to obtain isoform selectivity for JNK2, and it can be understood that there are not many inhibitors that have overcome this obstacle yet.

Regarding the structures of selective JNK3 inhibitors that are also selective for JNK2, all of them were observed to exhibit additional binding in the solvent-exposed moiety. A selective JNK3 inhibitor, namely compound **1** (Figure 9), forms an H-bond between protonated pyrrolidine and Asn89, and another selective JNK3 inhibitor, namely compound **3**, also forms an F-bond between the fluorine substituted in the indoline-2-one group and Asn152/Gln155. These bonds are all additional interactions in the solvent exposure area. Furthermore, the carbonyl group of thiazole-4-one in compound **3** was close to the Asn152 residue, which was predicted as a dipole interaction.

Once the isoform selectivity is achieved, there are still many obstacles to overcome. First of all, due to the similarities of the ATP-binding pockets that most protein kinase inhibitors target, general selectivity over a protein kinome other than JNK1/2 should be preferentially obtained. Second, a high potency will be required because the body concentration of therapeutics must compete with ATP, which is present in the millimolar range in cells. Thirdly, BBB penetration without the generation of major side effects will be required, which is the most difficult challenge since AD, the disease we are trying to conquer, is a chronic disease, and the therapeutics are required for long-term treatment. Therefore, in indications with such a long dosing period, a wide safety profile becomes a very important issue. Although the development of JNK3 inhibitors as a treatment for neurodegenerative disease has not been successful yet, it is becoming clear that JNK3 is a promising therapeutic target. 

JNK3 is attracting more and more attention as a therapeutic target for AD. These characteristics should be considered in the design of new JNK3 inhibitors, which could facilitate the development of potent and selective JNK3 inhibitors as therapeutics for AD in the near future.

## Figures and Tables

**Figure 1 biomedicines-09-01431-f001:**
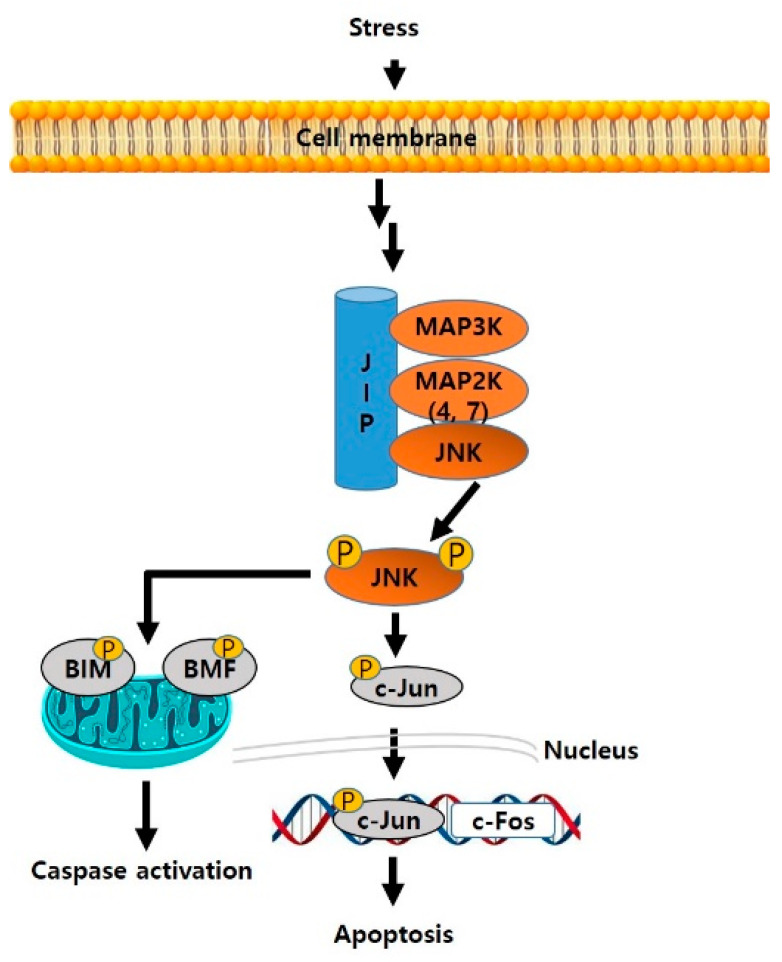
Overall JNK pathway from stress to apoptosis.

**Figure 2 biomedicines-09-01431-f002:**
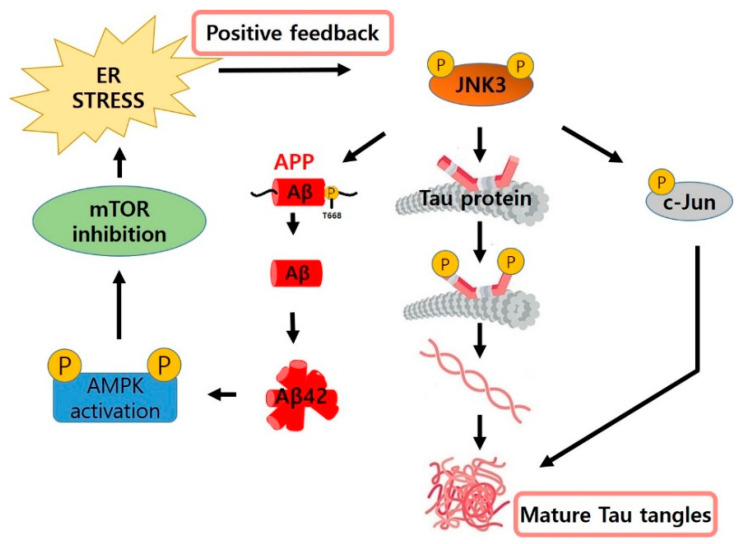
The role of JNK3 in the three pathological pathways of AD.

**Figure 3 biomedicines-09-01431-f003:**
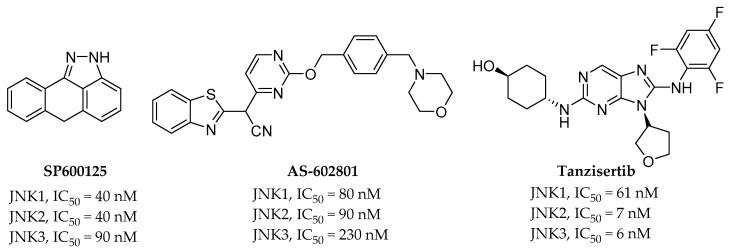
Chemical structures and activities of three representative pan-JNK inhibitors currently under development.

**Figure 4 biomedicines-09-01431-f004:**
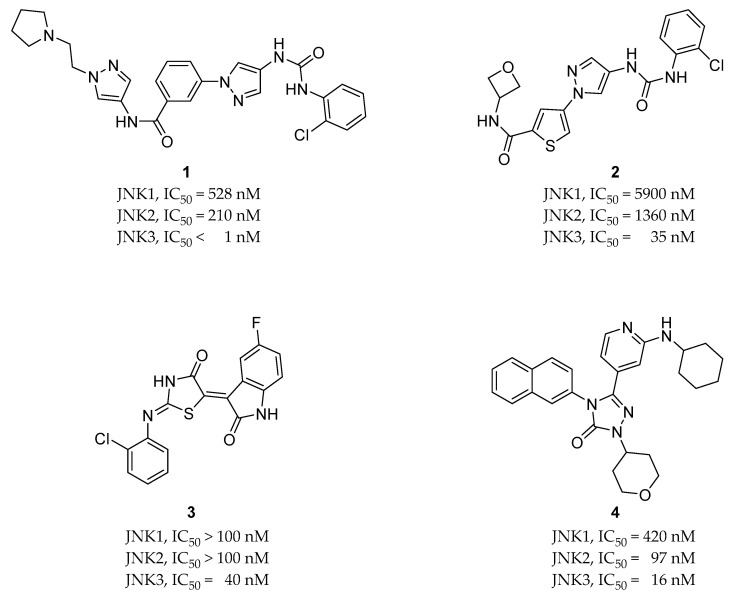
Chemical structures and activities of four JNK inhibitors with isoform selectivity that are currently under development.

**Figure 5 biomedicines-09-01431-f005:**
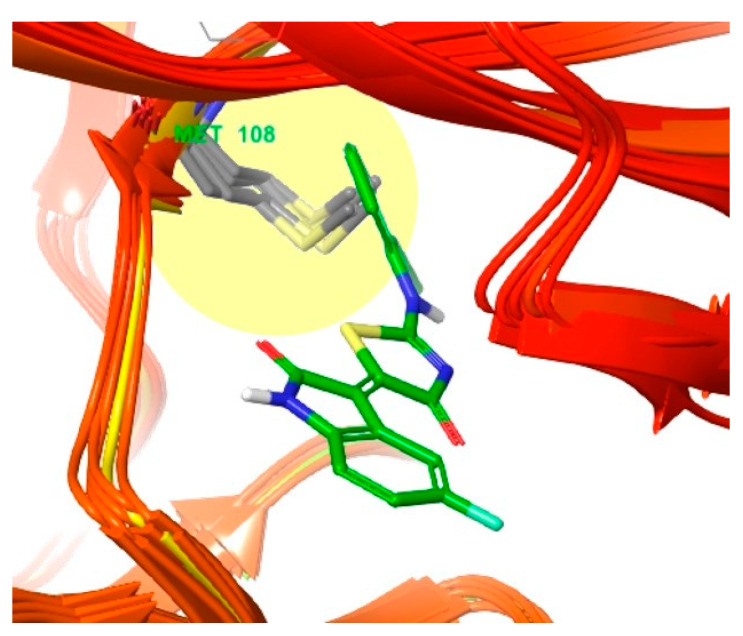
Superimposition of six crystal structures of JNK1 (PDB ID: 3PZE [26], 2XRW [27], 4QTD [28], 3ELJ [29], 4AWI [30], and 4L7F [31]) by pulling out the top six structures according to their resolution. The ligand is compound **3** docked into JNK3 (PDB 4WHZ [22]). Residue Met108 is shown in grey.

**Figure 6 biomedicines-09-01431-f006:**
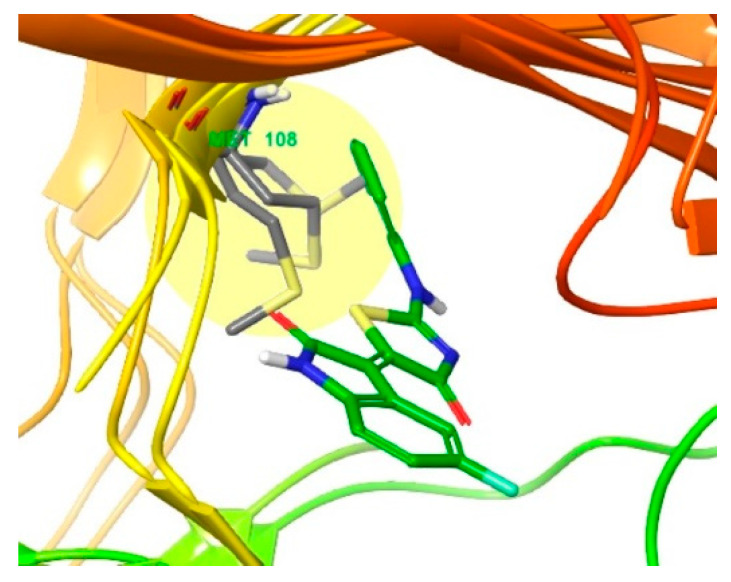
Superimposition of the crystal structures of JNK2 (PDB ID: 3E7O [32], 7CML [33], and 3NPC [34]), of which only three are published. The ligand is compound **3** docked into JNK3 (PDB 4WHZ [22]). Residue Met108 is shown in grey.

**Figure 7 biomedicines-09-01431-f007:**
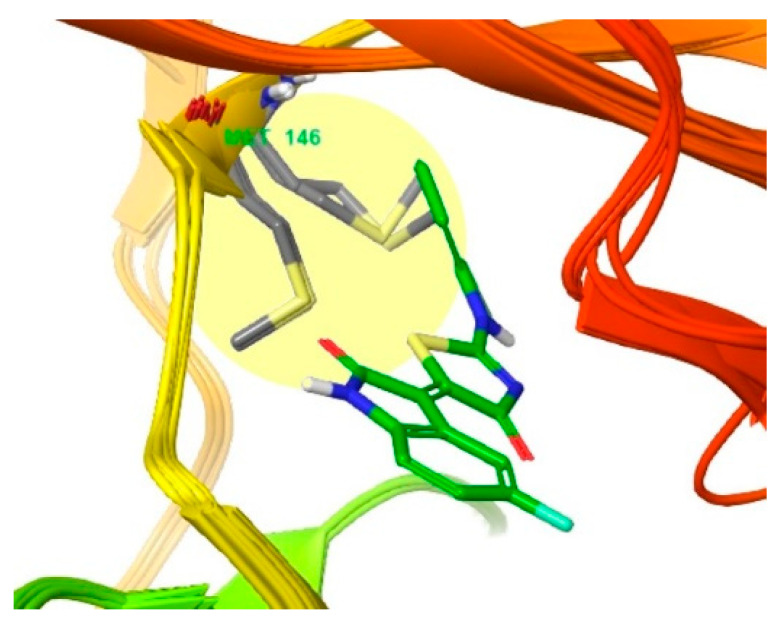
Superimposition of six co-crystals of JNK3 selected according to resolution (PDB ID: 4WHZ [22], 4W4W [35], 3OY1 [25], 7KSI [23], 6EMH [36], and 6EQ9 [36]). Compound **3** is docked into the active site of JNK3 (PDB ID: 4WHZ [22]). Residue Met146 is shown in grey.

**Figure 8 biomedicines-09-01431-f008:**
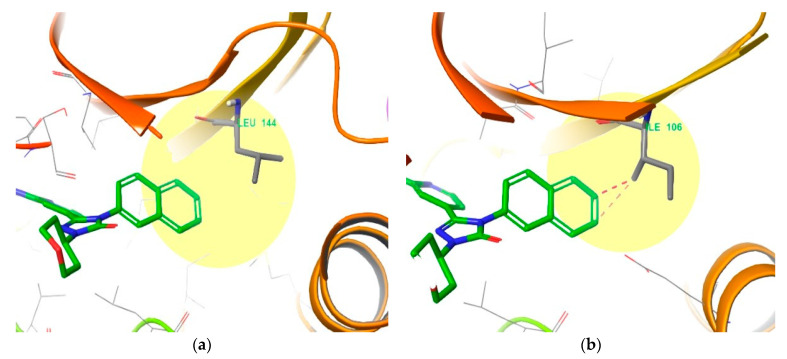
(**a**) Leu144 residue in the crystal structure of JNK3 co-crystalized with compound **4** (PDB ID: 3OY1 [25]), and (**b**) superimposition of the Ile106 residue in the crystal structure of JNK1 (PDB ID: 3PZE [26]) and compound 4 co-crystalized with JNK3.

**Figure 9 biomedicines-09-01431-f009:**
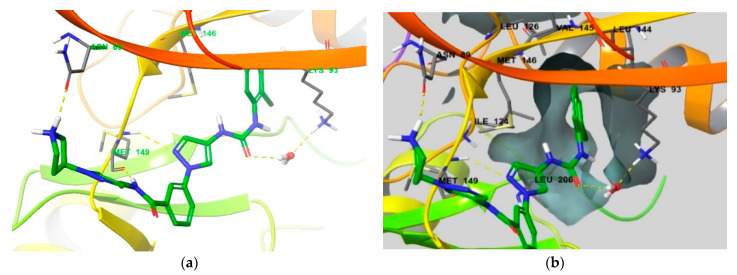
(**a**) Crystal structure of JNK3 bound to compound **1** (PDB ID: 4WHZ [22]). (**b**) Protein surface of the hydrophobic pocket is shown in the same JNK3 co-crystal. Residues that interact with compound **1** and comprise selectivity pocket are emphasized in the thin tube.

**Figure 10 biomedicines-09-01431-f010:**
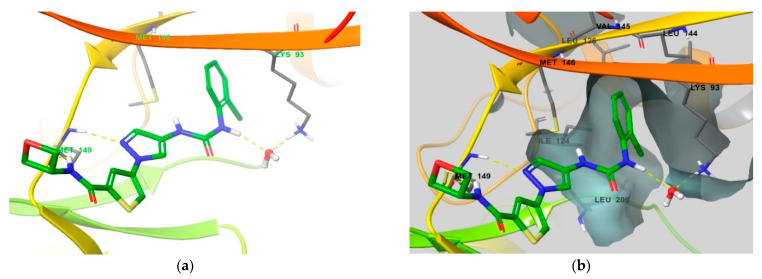
(**a**) Crystal structure of JNK3 bound to compound **2** (PDB ID: 7KSK [23]) (**b**) Protein surface of the hydrophobic pocket is shown at the same co-crystal. Residues that interact with compound **2** and comprise selectivity pocket are emphasized in thin tube.

**Figure 11 biomedicines-09-01431-f011:**
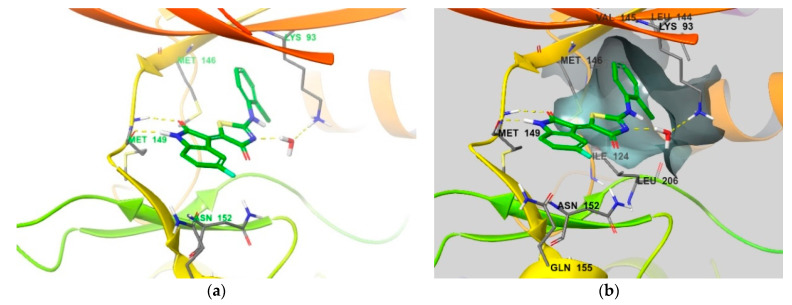
(**a**) Binding mode of compound **3** docked into the structure of JNK3 (PDB ID: 4WHZ [22]). (**b**) Surface of the hydrophobic pocket in active site is shown at the same docking structure of compound **3**. Residues that interact with compound **3** and comprise the selectivity pocket are emphasized in the thin tube.

**Figure 12 biomedicines-09-01431-f012:**
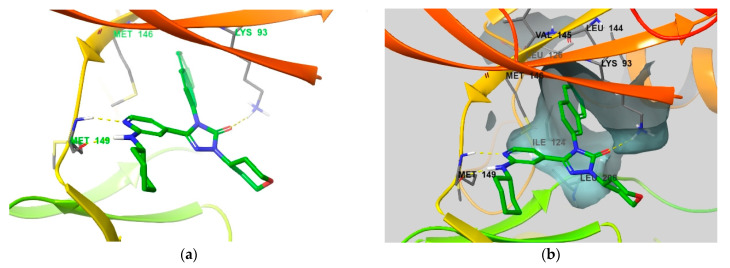
(**a**) Binding mode of compound **4** co-crystallized with JNK3 (PDB ID: 3OY1 [25]). (**b**) Surface of the selectivity pocket in active site is shown in the same co-crystal. Residues that interact with compound **4** and comprise selectivity pocket are emphasized in the thin tube.

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
