# Peer review of "A Perspective on the Development of c-Jun N-terminal Kinase Inhibitors as Therapeutics for Alzheimer’s Disease: Investigating Structure through Docking Studies"

_biomedicines, 2021, doi:10.3390/biomedicines9101431_

Round 1
Reviewer 1 Report
Protein kinases constitute a large group of enzymes that catalyze protein phosphorylation and control multiple signaling events. c-Jun-NH2-terminal-kinase (JNK) is a protein kinase, which induces transactivation of c-jun. There are three isoforms of JNK, JNK1, JNK2, and JNK3, which are encoded by three distinct genes. JNK1 and JNK2 are expressed ubiquitously throughout the body while the expression of JNK3 is limited and observed mainly in the brain, heart, and testes. JNK3 responds to pathophysiological events, such as stress response or cell death including apoptosis, and also regulates the physiological functions of neurons and non-neuronal cells, such as development, regeneration, and differentiation/reprogramming. JNK3 not only enhances amyloid β production, but also plays a key role in the maturation and development of neurofibrillary tangles. This review aims to screen for novel and selective inhibitors of JNK signaling for use as therapeutics in Alzheimer’s disease (AD) using a review of literature for past decade (2011-2021). This review particularly reviews studies focusing on structural perspective and docking insights of such inhibitors.
This review assembles information on JNK3 inhibitors studied in recent years, which could be helpful in developing therapeutics for AD.
My only concern is that this review needs editing for English language, and there are some errors, e.g. page 6 and page 12.
Author Response
Thank you so much for the result of the manuscript (Biomedicines-1403382). As you suggested "English Editing" has been processed and certificate is also attached. The corrections are colored in blue text.

Reviewer 2 Report
The manuscript "A perspective on the development of c-jun N-terminal kinase inhibitors as therapeutics for Alzheimer´s disease:investigating structure through docking studies" is a complet review of JNK inhibitors. The structural analysis of JNK active site gives interesting information about this inhibitors and may be useful for the development of new inhibitors for AD treatment. The manuscript is interesting and well organized.
Reviewer 3 Report
Protein kinases constitute a large group of enzymes that catalyze protein phosphorylation and control multiple signaling events. c-Jun-NH2-terminal-kinase (JNK) is a protein kinase, which induces transactivation of c-jun. There are three isoforms of JNK, JNK1, JNK2, and JNK3, which are encoded by three distinct genes. JNK1 and JNK2 are expressed ubiquitously throughout the body while the expression of JNK3 is limited and observed mainly in the brain, heart, and testes. JNK3 responds to pathophysiological events, such as stress response or cell death including apoptosis, and also regulates the physiological functions of neurons and non-neuronal cells, such as development, regeneration, and differentiation/reprogramming. JNK3 not only enhances amyloid β production, but also plays a key role in the maturation and development of neurofibrillary tangles. This review aims to screen for novel and selective inhibitors of JNK signaling for use as therapeutics in Alzheimer’s disease (AD) using a review of literature for past decade (2011-2021). This review particularly reviews studies focusing on structural perspective and docking insights of such inhibitors.
This review assembles information on JNK3 inhibitors studied in recent years, which could be helpful in developing therapeutics for AD.
My only concern is that this review needs editing for English language, and there are some errors, e.g. page 6 and page 12.
Author Response

(The authors gave the same response as above.)
